**communications**

**biology**

# Depth-dependent parental effects create invisible barriers to coral dispersal

Tom Shlesinger [1,2✉] & Yossi Loya[1]

Historically, marine populations were considered to be interconnected across large geographic regions due to the lack of apparent physical barriers to dispersal, coupled with a potentially widely dispersive pelagic larval stage. Recent studies, however, are providing increasing evidence of small-scale genetic segregation of populations across habitats and depths, separated in some cases by only a few dozen meters. Here, we performed a series of ex-situ and in-situ experiments using coral larvae of three brooding species from contrasting shallow- and deep-water reef habitats, and show that their settlement success, habitat choices, and subsequent survival are substantially influenced by parental effects in a habitat-dependent manner. Generally, larvae originating from deep-water corals, which experience less variable conditions, expressed more specific responses than shallow-water larvae, with a higher settlement success in simulated parental-habitat conditions. Survival of juvenile corals experimentally translocated to the sea was significantly lower when not at parental depths. We conclude that local adaptations and parental effects alongside larval selectivity and phenotype-environment mismatches combine to create invisible semipermeable barriers to coral dispersal and connectivity, leading to habitat-dependent population segregation.

[1] School of Zoology, The George S. Wise Faculty of Life Sciences, Tel-Aviv University, 69978 Tel-Aviv, Israel. [2] Present address: Institute for Global Ecology, Florida Institute of Technology, Melbourne 32901 FL, USA. ✉email: tomshlez@gmail.com

About 50 years ago, Janzen[1] examined how topographic variation in climate may shape an organism's physiological tolerances, creating barriers to dispersal and subsequent isolation of populations. He hypothesized that the lower amplitudes of temperature in the tropics, compared with temperate regions, would drive selection for organisms possessing narrow tolerances. Consequently, mountain passes in the tropics might represent a greater barrier to dispersal than similar altitude passes in temperate regions, since organisms would be more likely to encounter unfamiliar temperatures to which they were not adapted. This invisible barrier reflects the concept that greater sensitivity to environmental fluctuations is promoted by less frequent contact with such dynamics. Ample evidence from terrestrial ecosystems supports some of Janzen's predictions[2]. In marine ecosystems, however, research into similar invisible barriers has lagged behind and, although these barriers have been recognized[3–5], it is only relatively recently that such studies have gained momentum[6–12]. An emerging pattern of coral thermal tolerance related both to site conditions and to a species' thermal history can serve here as an example[13–22]. For instance, in some "depth-generalist" species, corals residing in deep reefs (30–75 m), termed mesophotic reefs, have acclimated to these deeper and cooler habitats and demonstrate lower thermal thresholds than their shallow conspecifics[19]. Their ability to colonize shallower reefs, which are exposed to higher temperatures, thus appears to be limited. In other words, corals at the mesophotic depths might be subject to invisible barriers to dispersal.

In many animals, the environment experienced by adults influences the phenotype and fitness of their offspring (i.e., parental effects)[23–26], and might impose additional invisible barriers to dispersal. Adult corals are attached to the seafloor and, like many other marine taxa, they disperse primarily through the production of planktonic larvae, which may drift and settle in environmental conditions that greatly differ from their parental environment. These larvae may disperse to environments in which they are less fit, and thereby experience higher non-random mortality compared with individuals that originated and settled locally. Several studies to date have indicated that parental effects influence the characteristics of coral oocytes, larvae, and subsequent juvenile fitness[12,18,27–31]. Specifically, there is some evidence pointing to higher larval settlement success in conditions that resemble those at the respective depths of the parent colonies[32–37]. Phenotype-environment mismatches (or "immigrant inviability"[38]) may therefore constitute a considerable barrier to population connectivity in the sea[8,39–41]. Indeed, recent studies on coral population genetic structures have found evidence of segregation across depth[10,42–50] (although this may differ between sites among species[46–49]). These studies motivated us to seek an explanation for the seemingly paradoxical findings of genetic differentiation in populations separated vertically by only tens of meters, alongside the findings of large-scale horizontal genetic connectivity, sometimes across hundreds of kilometers[10,42–50].

Furthermore, intermediate and mesophotic depths have been suggested as zones that may provide natural "refuges" from certain stressors, such as increasing ocean temperature, and that deeper reefs may play an important role in the replenishment of degraded shallow reefs by constituting a larval source[51,52]. One of the clearest examples of the refuge potential of mesophotic depths is found in the case study of the coral *Seriatopora hystrix* in Okinawa, Japan. Following a mass bleaching event in 1998, this species was reported to have become locally extinct[53,54]. However, recent studies performed in the same area have found abundant populations of this species in mesophotic depths[55,56], demonstrating the potential of deeper habitats to serve as refuges. Nonetheless, almost two decades after the initial disappearance of the species from the shallower depths, it is yet to be reported as having reestablished on the shallow reefs. Considering that planktonic coral larvae can be transported by ocean currents across large distances and considerable depths[57–60], this example raises the question: What might be preventing the mesophotic-depth population from reseeding the nearby shallow reef?

Here, we explored how parental effects, larval selectivity, and phenotype-environment mismatches might explain diverging coral population and community structures across depth. We performed a series of ex-situ and in-situ experiments, using a total of 4200 planulae (i.e., coral larvae) belonging to three brooding species from contrasting shallow- and deep-reef (i.e., mesophotic) habitats, to investigate how parental habitat conditions affect planulae settlement rates, settlement choices, survivorship, and growth. Planulae from different coral species and depths were subjected to two kinds of experiments: (1) "no-choice" experiments—planulae were divided among several light regimes and settlement-substrate treatment groups; and (2) "choice" experiments in which planulae were provided with two settlement substrates (tiles); one that had been conditioned in a shallow habitat (~5 m) and another that had been conditioned in a mesophotic habitat (~45 m). At the end of these experiments the tiles containing newly settled corals, which originated from known depths, were divided into two groups and translocated to the two contrasting habitats (i.e., to the shallow and mesophotic depths) to assess their survival and growth in the different habitats. Our results show that parental effects and biological responses, during the early life-history stages of brooding corals, impose selective constraints on their dispersal and settlement under different conditions, depending on their natal habitats. Thus, invisible barriers to recruitment may structure coral populations in a habitat- or niche-dependent manner related to their parental environment.

## Results and Discussion

**Parental effects influence settlement**. Possible parental effects were initially assessed by comparing the settlement success of planulae originating from the two extremes of the species' depth range (Fig. 1) under controlled environmental manipulations. Planulae of corals from 5 and 45 m depth were collected from two "depth-generalist" species, *Rhytisma fulvum* and *Stylophora pistillata*, and planulae of a "deep-specialist" species, *Stylophora kuehlmanni*, were collected from colonies occurring at 45 m depth. The planulae were then divided among four treatment groups simulating different combinations of environmental conditions (i.e., light regime and settlement substrate) at the shallow (~5 m) and mesophotic (~45 m) depths. We counted the settled planulae daily for a period of 10 days and found that those obtained from deep-water populations demonstrated a more treatment-specific response (Fig. 2a–e).

Planulae of the "deep-specialist" coral, *S. kuehlmanni*, showed significant differences in settlement between the light regimes (permutation repeated-measures ANOVA, $p < 0.001$, Cohen's $d = 2.55$), but not between the settlement tiles. At the end of the experiment, under the light regime simulating mesophotic-depth conditions, the percentage of settlement was more than double the percentage under the shallow-depth light regime (Fig. 2a). By contrast, the planulae of *S. pistillata* colonies from shallow water did not show any differences in settlement between the different treatments (Fig. 2b, c). The mesophotic-originating planulae of *R. fulvum* were significantly different in settlement between the two types of settlement tiles (permutation repeated-measures ANOVA, $p < 0.01$, Cohen's $d = 0.9$), but not between the light regimes (Fig. 2d). By contrast, shallow-originating planulae of this species revealed no significant differences

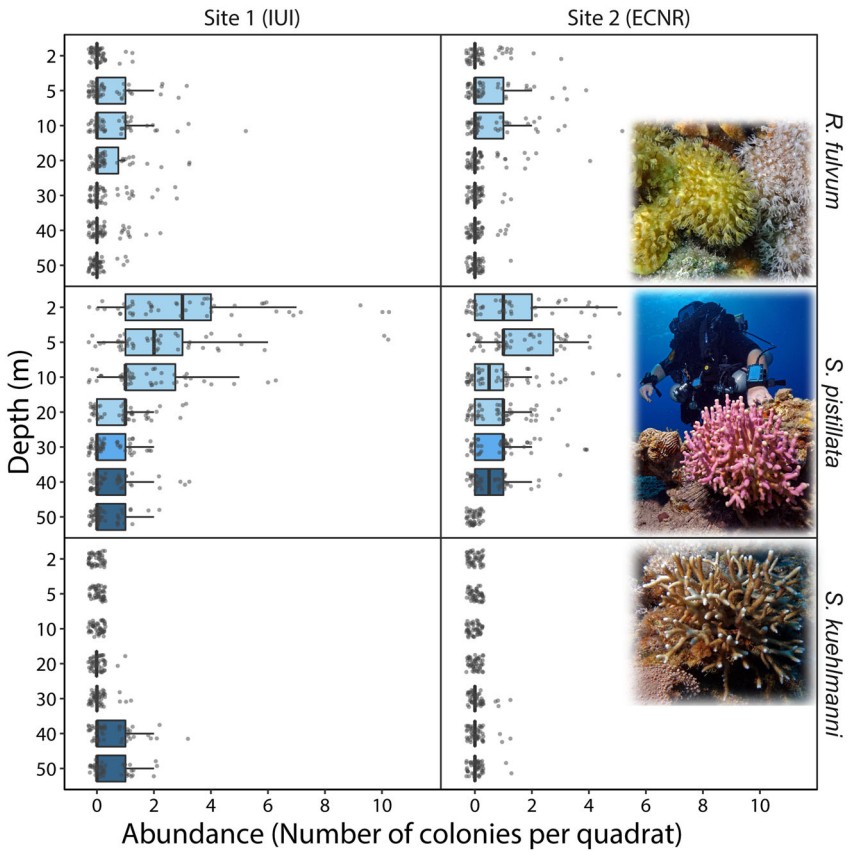

**Fig. 1 Abundance of the three focal species at the two study sites along a 2–50 m depth gradient.** The abundance is presented as the number of colonies per quadrat via box plots, with center lines indicating the medians, boxes indicating the lower and upper quartiles, and whiskers indicating 1.5x interquartile range. Points represent individual quadrat counts ($n = 50$ for each site and depth combination).

between either light regimes or settlement tiles (Fig. 2e). Daily, seasonal or annual amplitudes of several environmental factors at the mesophotic depths may be smaller than those at the shallow depths[19,61]. For example, in our study site, the temperature fluctuations and maximum values at shallow depths are larger than at mesophotic depths[62]. Additionally, as light attenuates rapidly as it travels in the sea, both its amplitudes and maximum intensity are much larger at shallow depths than at mesophotic depths[61,63]. Hence, our findings that mesophotic-originating planulae may have more constraints on their settlement and survival (i.e., narrower tolerances or niches) corroborate the contention that exposure to more variable conditions promotes tolerance to a wider range of environmental conditions[1].

Our results further indicate that in brooding coral species inhabiting a wide depth gradient, the directionality of larval supply may be more prominent from shallow to deep waters rather than vice versa. These results parallel several studies presenting genetic evidence of asymmetric (shallow-to-deep) gene-flow across depth[10,47,49]. Additionally, although total coral cover and species composition differed significantly between the two sites surveyed in this study[62,64], the abundance of the two "depth-generalist" species revealed a similar pattern that might indicate their natural preferences (Fig. 1). Although these species are relatively common at the mesophotic depths, their abundance at the shallow depths (~2–10 m) outnumbered their abundance at mesophotic reefs. Taken together with the possibly narrower thermal tolerance[19] and reduced fertility and reproductive performance at the mesophotic depths[65], deep-water populations of some "depth-generalist" species may reflect marginal populations, which may be dependent to some extent on larval supply

from shallower populations. These, in turn, may indicate that deep-water populations might be more fragile than commonly assumed, and thus warrant immediate conservation considerations in their own right[66–68]. Nonetheless, in regions where shallow reefs are already severely degraded, the potential reproductive output and subsequent larval success of locally intermediate- or mesophotic-depth populations may exceed that of the shallow-reef populations[29,69].

**Neonate coral settlement choice.** Coral larvae appear to be able to detect and discriminate between a range of chemical cues, acting as either negative or positive cues to the induction of larval settlement and metamorphosis[70]. Such cues are generated by a variety of benthic or fouling organisms such as calcareous algae, microbial biofilms, and more[70–72]. To further assess whether parental effects might influence planulae behavior and preferences, we performed "choice" experiments in which we offered the planulae two different settlement tiles (shallow- and mesophotic-conditioned) simultaneously (Fig. 2f–h). Given the choice of two tiles, planulae tended to settle more commonly on the tile reflecting their parental habitat. While mesophotic-originating planulae of R. fulvum showed a significant preference for settling on tiles conditioned in the mesophotic depth (permutation repeated-measures ANOVA, $p = 0.03$, Cohen's $d = 1.12$), there were no differences in percentage of settlement for the shallow-originating planulae of this species between the two types of tiles (Fig, 2f, g). Shallow-originating planulae of S. pistillata, in contrast, revealed a striking significant preference for tiles conditioned in the shallow depth (permutation repeated-measures

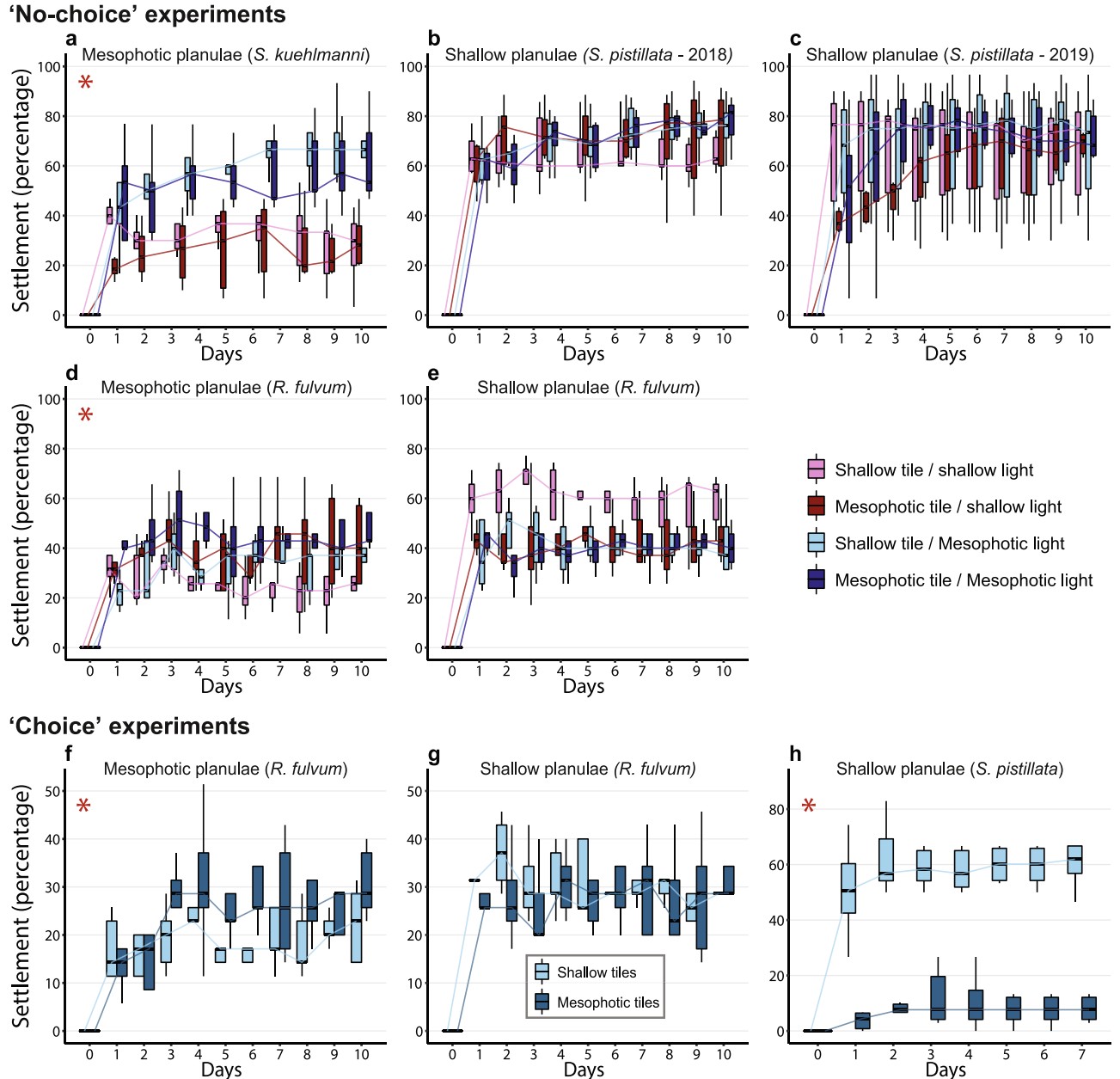

**Fig. 2 Settlement of coral larvae originating from different depths and species.** Box plots showing the settlement percentages in the "no-choice" (**a-e**) and "choice" (**f–h**) experiments (For every treatment group, $n = 5$–6 replicates containing 30–40 larvae each; for detailed account see Methods and Supplementary Data 1). Settlement percentages of **a** the "deep-specialist" species *Stylophora kuehlmanni*; **b** shallow-originating larvae of the "depth-generalist" species *Stylophora pistillata* in 2018; **c** shallow-originating larvae of *S. pistillata* in 2019; **d** mesophotic-originating larvae of the "depth-generalist" species *Rhytisma fulvum*; **e** shallow-originating larvae of *R. fulvum*; **f** mesophotic-originating larvae of *R. fulvum*; **g** shallow-originating larvae of *R. fulvum*; and, **h** shallow-originating larvae of *S. pistillata*. Center lines of the box plots indicate the medians, boxes indicate the lower and upper quartiles, and whiskers indicate 1.5x interquartile range. Lines connect the medians of each treatment group to aid visualization and red asterisks indicate where significant differences were found between treatments.

ANOVA, $p < 0.001$, Cohen's $d = 4.49$), with six times higher settlement percentage than on mesophotic-conditioned tiles (Fig. 2h). While we did not examine the taxonomic composition of fouling communities on the tiles, earlier studies indicated that they may differ considerably across depths[73], even within a smaller range than that of our study[35,71]. Thus, our results indicate that although larval settlement responses to substrate cues are not absolute, for example, mesophotic *R. fulvum* larvae also settled on shallow-conditioned tiles, their response to such cues can vary not only between species (Fig. 2g, h) but also

between larvae of the same species, originating from different habitats (Fig. 2f, g).

**Phenotype-environment mismatches.** The reciprocal-depth-translocation experiments of coral settlers revealed significantly higher growth rates in shallow water than in deep water regardless of parental origin, although the survival of settlers in their respective parental habitat was generally the highest (Fig. 3). In all experiments, the survival rates of translocated juvenile

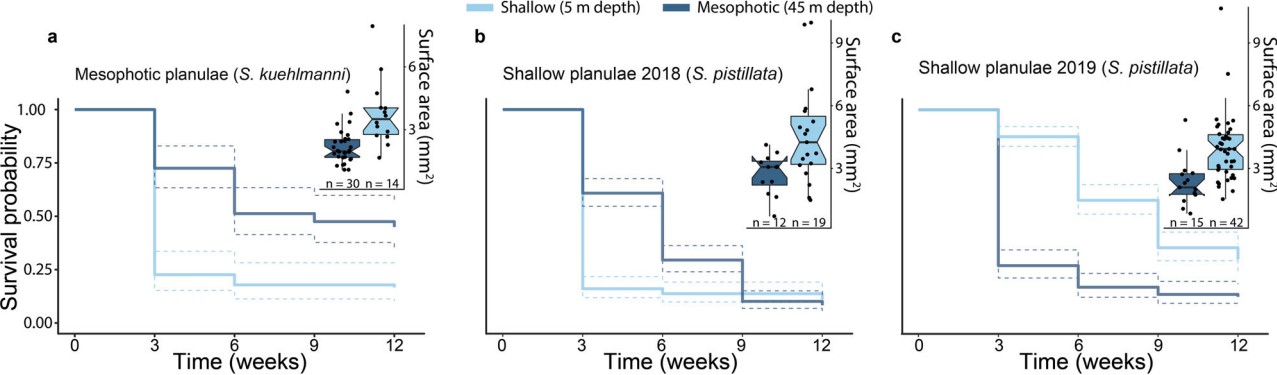

**Fig. 3 Survival and growth of juvenile coral settlers translocated to shallow and mesophotic depths.** Kaplan–Meier survival curves of coral settlers originated from **a** mesophotic-reef *Stylophora kuehlmanni* larvae ($n = 84$ and 80 for the shallow and mesophotic translocations, respectively); **b** shallow-reef *Stylophora pistillata* larvae in 2018 ($n = 218$ and 217 for the shallow and mesophotic translocations, respectively); and, **c** shallow-reef *S. pistillata* larvae in 2019 ($n = 198$ and 178 for the shallow and mesophotic translocations, respectively). Dashed lines indicate 95% confidence interval. Box plots in each panel show the size of the surviving corals at the end of each experiment (as the data points), with center lines indicating the medians, boxes indicating the lower and upper quartiles, and whiskers indicating 1.5x interquartile range. Significant differences in the survivorship and growth between depths were found in all experiments.

corals differed significantly between the two different depths (log-rank test with Bonferroni correction following an estimation of Kaplan–Meier survival curves, $P < 0.001$ for all). The survival rate of juvenile corals derived from mesophotic-reef planulae of *S. kuehlmanni* and translocated to the shallow depth (Fig. 3a) declined sharply within the first census period (i.e., after 3 weeks), with only a minor further decline by the end of the experiment (i.e., after 3 months). By contrast, for juveniles translocated to the mesophotic depth the survival rate declined gradually and at the end of the experiment, it was almost three times higher than that in the shallow depth (Fig. 3a). Survival rates of juvenile settlers derived from shallow-reef planulae of *S. pistillata* differed between the same experiments performed in different years (Fig. 3b, c). In 2018, the survival rate at the shallow depth declined sharply within the first census period and had not changed much by the end of the experiment. At the mesophotic depth, the survival rate had a more gradual decline, although ending with a lower survival rate than at the shallow depth (Fig. 3b). In 2019, by contrast, the survival rate at the mesophotic depth showed a sharp decline within the first census period and remained relatively steady until the end of the experiment; whereas at the shallow depth the survival rate declined more gradually, ending more than two-fold higher than at the mesophotic depth (Fig. 3c). At the end of all the translocation experiments (i.e., 3 months), the size of the surviving corals was significantly larger at the shallow depth than at the mesophotic depth (permutation two-way ANOVA, $p < 0.001$, Cohen's $d = 1.18$) and did not differ between species or between years (box plots in Fig. 3). The different life-history stages of *S. pistillata* and *R. fulvum* are depicted in Fig. 4a–d and 4e–h, respectively. At the end of the experiments, corals that had been translocated to the shallow depth were almost twice the size of those translocated to the mesophotic depth, regardless of parental origin of the planulae (Figs. 3 and 4c, d).

The results of the settlement experiments, together with the differential survival of settlers in-situ indicate that biological responses and parental effects may impose selective constraints on larval dispersal, settlement, and survival. These adds to the accumulating evidence of a more specific or local larval retention within reefs[59,74–78], suggesting that populations of some marine species may be less "open"[59] than traditionally believed and are structured, to some extent, in a niche- or habitat-dependent

manner related to their natal environment. Importantly, although we have investigated two main environmental factors (i.e., light regime and settlement substrate), there might be additional factors that further constrain dispersal and connectivity. For example, temperature differences are likely to be another conspicuous factor as both the range and amplitude of temperatures can differ considerably between depths in different localities[61]. While we were able to avoid confounding effects of temperature in this study by performing most of the settlement experiments when the temperatures at shallow and mesophotic depths are similar (Fig. 5a, b), and by having our experimental aquaria supplied with water from ca. 40 m depth, further research focusing on temperature effects and additional environmental factors are warranted.

Unfortunately, the mesophotic-depth colonies of the "depth-generalist" species, *S. pistillata*, did not appear to release planulae during our experiments period, except for few colonies that released very few planulae, so we were not able to perform a fully-crossed experiment with this species. To qualitatively assess these colonies' reproductive state, we dissected a small sample of each colony and found that most of them were devoid of or contained very few gametes or planulae. This result supports similar reports on reductions in fecundity or oocyte and larvae size with depth found in several species in the region[62,79,80]. Nonetheless, the overall results from our fully-crossed experiments with the "depth-generalist", *R. fulvum*, together with the results of shallow-originating planulae of *S. pistillata* and mesophotic-originating planulae of the "depth-specialist", *S. kuehlmanni*, demonstrate the possible invisible barriers imposed on larval dispersal of corals, at least for brooding species.

While there is some potential for long competency periods and subsequent long-distance dispersal of larvae of brooding corals[57], they are generally believed to have more localized dispersal patterns than those of broadcast-spawning corals[49,65,70,81]. However, larval settlement of spawning species might occur earlier than commonly assumed[82] and, therefore, their dispersal might also be restricted to some extent. Moreover, larvae of spawning corals were also found to have differential susceptibility to light as reflected by different levels of photo-protective compounds, settlement success, and survival, depending on their parental depths[32–34,37]. Nonetheless, it is commonly hypothesized that brooding corals' populations are less connected and that

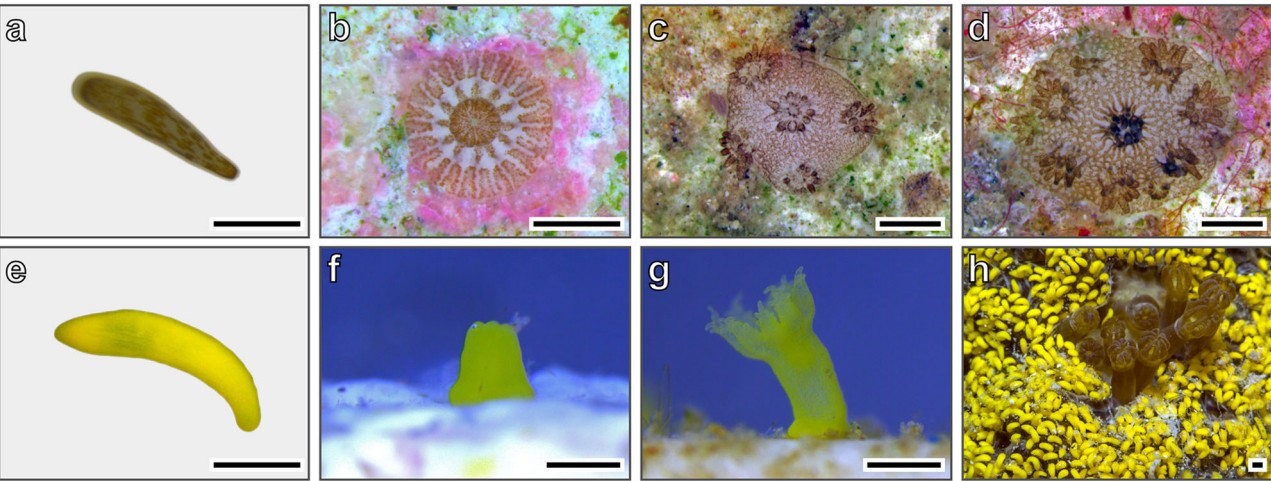

**Fig. 4 Early life-history stages of the coral species studied. a** *Stylophora pistillata* free-swimming larva. **b** A 2-day-old primary polyp of *S. pistillata*. **c** A 3-month-old *S. pistillata* originating from shallow-reef larva and translocated to the mesophotic depth. **d** A 3-month-old *S. pistillata* originating from shallow-reef larva and translocated to the shallow depth. **e** *Rhytisma fulvum* free-swimming larva. **f** Shortly after settlement, a *R. fulvum* larva metamorphosing into a primary polyp. **g** A 10-day-old polyp of *R. fulvum*. **h** Surface-brooding colony of *R. fulvum*. Scale bars indicate 1 mm.

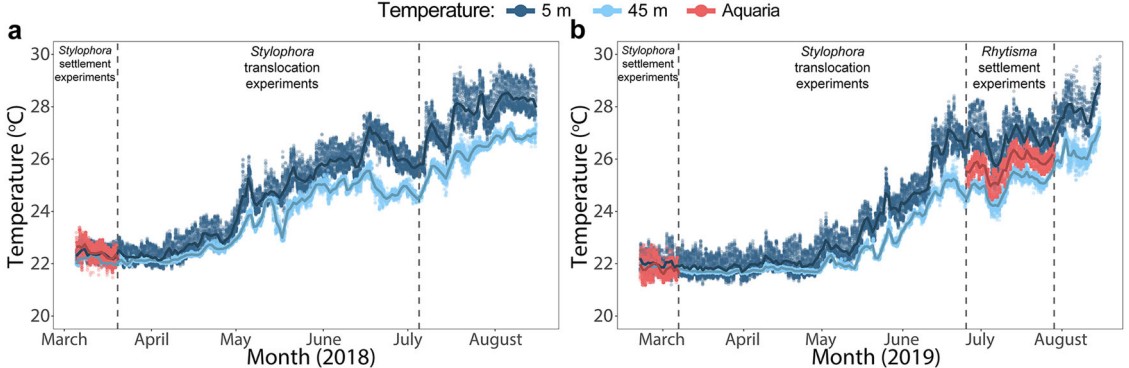

**Fig. 5 Temperature regime during the study period.** Temperatures in the experimental aquaria and in the open sea at depths of 5 and 45 m during (**a**) 2018 and (**b**) 2019. Points represent continuous measurements taken every 15 min and lines represent the daily temperature means.

divergence by depth might be more prevalent in brooders than in spawners[49]. In some species and localities, divergence of coral populations by depth might be further reinforced by specialized depth-specific coral host–symbiont associations[44,46,48,83,84], which may also be related to the mode of larval development. Larvae of most brooding coral species directly inherit their dinoflagellate photosymbionts from their parent colonies (i.e., vertical transmission), while larvae or newly established polyps of many spawning corals acquire symbionts from the environment (i.e., horizontal transmission)[27,85]. Thus, the parental effects we report here likely involve the mutual response of both the coral host and its endosymbiotic assemblage, which may further suggest that brooding corals predispose the symbiotic association of their offspring to optimal performance under parental habitat conditions. Whether our findings generally hold for corals as well as for other marine organisms, awaits similar assessments of broadcast-spawning species to be carried out. Yet, despite the natural variability between species and sites, genetic segregation of both the coral host and its symbionts across depth was found to date in both brooding and spawning corals[10,42–50,83,84]. Regardless, our results demonstrate that although not fully hindered, corals' early life-history stages may have differential success and fitness in different conditions, depending on their natal habitats.

## Conclusions

With increasing threats[86–88] and major declines in coral reefs worldwide[22,86,87], a fundamental understanding of how patterns of larval dispersal and connectivity emerge and ultimately create population and community structures is essential. Depth-related divergence seems to drive differentiation among closely-related species[3,5,9]. Thus, depth may play a considerable role in ecological speciation, which refers to the evolution of reproductive isolation between populations, or to subsets of a single population by adaptation to different habitats or to ecological niches[6,89,90]. The parental effects and phenotype-environment mismatches that we found here in brooding corals reinforce the notion that long pre-reproductive selection, combined with possible assortative mating, may facilitate the isolation and divergence of coral populations across differing habitats[10,42,49,62]. At the community level, coral reefs have long been recognized as displaying a distinct depth zonation, dominated by different species[91–97]. At the population level, recent studies have also found evidence of ecological adaptive divergence and genetic segregation across depth in several species and localities[10,42–50,83,84]. Here, we have shown that local adaptations and parental effects may considerably influence brooding coral abundance across depth through selective larval settlement and phenotype-environment mismatches. In combination, these processes result in habitat-

dependent recruitment and survival, and thereby create invisible semipermeable barriers to coral dispersal and connectivity, which may explain some of the observed community and population zonation across depth.

Moreover, our findings may also explain the somewhat counter-intuitive patterns of large-scale horizontal genetic connectivity across tens or hundreds of kilometers, alongside genetic differentiation of populations separated vertically by only a few tens of meters. Accordingly, while similar studies using broadcast-spawning species are needed to assess the generality of the results we report here, our findings highlight that the connectivity and patterns of population and community structures of many marine taxa are not only be the simple consequence of ocean currents and larval duration in the plankton but may be also substantially shaped by biological responses in their early life-history stages.

## Methods

**Studied species**. We used three larvae-brooding coral species: two scleractinian corals (i.e., stony corals) *S. pistillata* and *S. kuehlmanni*, and an octocoral (i.e., soft coral) *R. fulvum*. Both *S. pistillata* and *R. fulvum* are "depth-generalists", abundant from the reef flat and down to ca. 50 m depth, and they are both common throughout the entire Indo-Pacific region. The third species, *S. kuehlmanni*, is a "deep-specialist", mostly abundant at depths of 40–60 m and only rarely found at depths shallower than 25–30 m. It is an endemic species to the Red Sea and to parts of the western Indian Ocean adjacent to the Red Sea. Both the *Stylophora* species are hermaphroditic brooders, while *R. fulvum* is a gonochoric surface-brooder; i.e., following the release of sperm from male colonies, the oocytes are fertilized within the female colonies and the developing planulae are then brooded for 6 days on the external surface of the colonies[79,98]. *S. pistillata* is planulating throughout at least 8 months, and peaks in the number of colonies releasing planulae between February and May[99]. The depth distribution and abundance of the three studied species were estimated by counting all the individuals in $50 \times 70$ cm photo-quadrats imaged during our previous study[62] at two sites along a depth gradient of 2–50 m ($n = 50$ for each site and depth combination).

**Coral and larvae collections**. On March 4, 2018, we collected five colonies of *S. pistillata* from both shallow (~5 m) and mesophotic (~45 m) depths and five colonies of *S. kuehlmanni* from the mesophotic depth (~45 m). We collected corals on March since (i) this timing denotes the peak planulation period[99], and (ii) at this time and at the translocation experiments period that followed, we could maximally avoid confounded effects of major temperature differences between shallow and mesophotic depths in both years (2018 in Fig. 5a and 2019 in Fig. 5b). To monitor water temperature, data loggers (HOBO Water Temp Pro v2, Onset Computer Corporation) set to take measurements every 15 min were placed in the experimental aquaria and additional ones were fixed to the reef at 5 and 45 m depth. Following collection of the corals, each colony was placed in an individual tank with flowing seawater and equipped with a small container featuring 120 μm mesh sides at the water outflow of the tank to collect the planulae (Fig. 4a, e). These tanks were located in an outdoor facility under ambient conditions and thus were exposed to natural photoperiods (ca. 12 h in March and 15 h in July). Additionally, constant water supply to the facility from ca. 40 m depth aid in preventing water temperature from reaching the high values characterizing shallow depths during the warm summer months (Fig. 5). Moreover, the tanks were covered with neutral density and "deep blue" filters (LEE Filters, UK) to simulate the light regime at shallow (~3–5 m) and mesophotic (~40–50 m) depth, respectively. Ambient light levels in the region range between 1290 μmol m$^{-2}$s$^{-1}$ in the winter and 2210 μmol m$^{-2}$s$^{-1}$ in the summer[63], and as measured by Rosenberg et al.[100], the neutral density filters reduced light levels from ca. 1970 μmol m$^{-2}$s$^{-1}$ to 1580 μmol m$^{-2}$s$^{-1}$, making it comparable to ca. 3–5 m depth[63,100,101]. Light intensity in aquariums with "deep blue" filters were measured by Eyal et al.[101] and ranged between 23 μmol m$^{-2}$s$^{-1}$ in the winter and 118 μmol m$^{-2}$s$^{-1}$ in the summer, making it comparable to ca. 40–50 m depth[63,101].

Except for the *S. pistillata* colonies from the mesophotic depth, within 2–3 nights all the other corals had released all the planulae necessary for the experiments. Colonies were kept in isolation for two additional weeks to monitor larval release, which was determined four times a day: about 30 min after sunrise (~06:30); in the afternoon (~16:00); about 30 min after sunset (~18:15); and, at 22:00. Over the 2-week period that *S. pistillata* colonies were held in aquaria for planulae collection, most of the mesophotic-reef colonies of this species did not release any planulae and some released only a few, which were insufficient for even one treatment group of any experiment. Consequently, we repeated the procedure in 2019 and carried out an additional collection of corals on February 20, 2019. However, similar to the previous year, the mesophotic-reef colonies of *S. pistillata* again released no or very few planulae throughout the 2 weeks in which they were

kept in the aquaria. We then preserved and dissected samples of these colonies to assess their reproductive state. Samples were fixed in 4% formaldehyde solution in seawater for 24 h, rinsed in running tap water, and preserved in 70% ethanol. Then, the samples were decalcified using a solution of 25% formic acid buffered with sodium citrate and dissected under a Nikon SMZ1500 stereo microscope. Planulae of the soft coral *R. fulvum* were collected in-situ from 7–10 colonies at both ~5 m and ~45 m depth on day 5 or 6 of the surface-brooding event (Fig. 4h), i.e., when the planulae are mature, elongated, and ready to detach from the colony surface and settle[79,98]. During this study, *R. fulvum* shallow population bred 3 weeks prior to the mesophotic population, as reported previously[79,98], and collections of planulae were made on June 25, 2019 for the shallow-reef colonies and on July 16, 2019 for the mesophotic-reef colonies. Prior to the initiation of each experiment, planulae from all colonies for each given depth and for each species were mixed and then randomly subsampled and distributed to the different treatments.

**Ex-situ settlement experiments**. To investigate coral parental effects, larval settlement selectivity, phenotype-environment mismatches, and dispersal potential between differing habitats (shallow and mesophotic), we carried out a series of experiments incorporating shallow- and mesophotic-originating planulae of the three studied species. To determine the parental effects of a coral on planulae fitness and selectivity, we manipulated two fundamental environmental variables, the light regime and settlement substrate, and performed two kinds of ex-situ experiments—"no-choice" and "choice" experiments. The light regimes were manipulated using the neutral density and "deep blue" filters described above in the same outdoor aquaria facility. To manipulate the settlement substrates, we used settlement tiles ($10 \times 10$ cm each) that were preconditioned for 5–6 months at their designated depth (5 or 45 m depth) prior to the onset of each experiment. We preconditioned these tiles for a long period (ca. half a year) to allow them to acquire the particular cover of calcareous algae, bacterial communities, etc. typical of each depth.

In the "no-choice" experiments, planulae were placed in glass beakers and divided among four treatments: (1) Shallow-reef light regime and settlement tiles preconditioned on a shallow reef; (2) Mesophotic-reef light regime and settlement tiles preconditioned on a shallow reef; (3) Shallow-reef light regime and settlement tiles preconditioned on a mesophotic reef; and (4) Mesophotic-reef light regime and settlement tiles preconditioned on a mesophotic reef. In the "choice" experiments, planulae were placed in glass beakers containing two settlement tiles: one that had been preconditioned on a shallow reef (ca. 5 m depth) and the other that had been preconditioned on a mesophotic reef (ca. 45 m depth). The choice experiments were performed in 2019, where we only collected the two species *S. pistillata* and *R. fulvum*, therefore, we did not perform a "choice" experiment with larvae of *S. kuehlmanni*. In all experiments, each treatment group comprised 5–6 replicates (i.e., beakers), each containing 30–40 planulae randomly subsampled from a pool of all released planulae from colonies of the same depth (see Supplementary Data 1 for exact numbers) and approximately 600 μl of filtered seawater. To maintain water quality within the beakers, we added ca. 200 μl of fresh filtered seawater daily and after 5 days the entire water volume in each beaker was replaced with fresh filtered seawater. Seawater was filtered daily using a flow-thru filter (Supor DCF capsule filter, 0.2 μm, Pall Corporation). Settled planulae (Fig. 4b, f, g) were counted daily for a period of 10 days using a magnifying glass. Thereafter, none or very few free-swimming planulae remained in each replication.

**In-situ survival and growth experiments**. Following the ex-situ experiments with the two *Stylophora* species planulae, the tiles, which held coral settlers from known depth origins, were translocated to the sea in a reciprocal manner (i.e., both shallow- and mesophotic-originating planulae to both shallow and mesophotic depths) in order to assess the possible consecutive phenotype-environment mismatches. To that end, the tiles were placed on plastic racks, translocated to the open sea, and secured to metal tables at 5 and 45 m depth. The racks were covered with a 2 cm aperture plastic mesh, which is large enough to allow both currents and light to travel through freely, while preventing predation and grazing by different animals. At 3-week intervals, for a period of 3 months, we collected the tiles and counted the number of survivors immediately upon their retrieval from the sea. Given the steep bathymetric slope in this area, we were able to retrieve both mesophotic and shallow racks within a short 15–20 min dive directly from the beach and placed them immediately after in the aquaria facility located on the beach, thus minimizing the exposure of the mesophotic tiles to shallow-water conditions. Within 1 day of collection all tiles were placed back in the sea. At the end of the experimental period, the size of the surviving corals was determined by measuring their surface area using a microscope equipped with a camera (Leica M165 FC and Leica DFC295, respectively).

**Statistics and Reproducibility**. Statistical analyses were made using R v3.6.1[102]. Mixed-effects (Repeated-measures) ANOVAs were used to assess differences in settlement between the different treatments. When significant differences were found, the effect sizes of either light regime or settlement tiles were estimated by calculating Cohen's *d* for the settlement percentages at the end of the experiment. Two-way ANOVA was used to assess differences in coral growth between depths and planulae origin. Since in most cases the data did not conform to the assumptions of parametric

tests (e.g., normality and homogeneity of variances), we used a permutation approach throughout. Survival curves and probabilities of the juvenile coral settlers translocated to the open sea were estimated by Kaplan–Meier survival analysis. Differences between the survival curves of planulae groups and translocated depths were sought by pairwise comparisons using log-rank test adjusted by Bonferroni correction. Sample sizes were as described for the individual experiments above.

**Reporting summary**. Further information on research design is available in the Nature Research Reporting Summary linked to this article.

## Data availability

All coral settlement experiments and surveys data generated and analyzed during this study are included in Supplementary Data 1.

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

## Acknowledgements

We thank Ronen Liberman, Dror Komet, Mark Chernihovsky, Or Ben-Zvi, Hanna Rapuano, Jessica Bellworthy, Genadi Zalzman, and Ofir Hameiri for assistance in fieldwork. We thank Rob van Woesik, Derya Akkaynak, and Dan Holstein for helpful reviews and comments on earlier versions of the manuscript. We would also like to thank the Underwater Observatory Marine Park and the Interuniversity Institute for Marine Sciences at Eilat for providing the infrastructure and support for the experiments. This research was supported by an Israel Science Foundation grant no. 1191/16 to Y.L. and fellowships from the Israeli Taxonomy Initiative, Interuniversity Institute for Marine Sciences in Eilat, Rieger foundation, and PADI foundation to T.S.

## Author contributions

T.S. and Y.L. conceived and designed the study. T.S. performed the fieldwork, experiments, data analyses, and led the writing of the manuscript, with both authors contributing to the final manuscript.

## Competing interests

The authors declare no competing interests.
