## [Peer Review File · Communications Biology]

This manuscript has been previously reviewed at another Nature Research journal. This document only contains reviewer comments and rebuttal letters for versions considered at Communications Biology.

REVIEWERS' COMMENTS:

Reviewer #3 (Remarks to the Author):

This is my second review of this manuscript, and the authors have made a careful set of revisions in response to my and the other reviewers' comments. As mentioned previously, while the central question of the study is not exactly novel - namely that habitat (in this case depth) structures the success of coral settlement and success, and that structuring may be reinforced through generational selection or adult fitness. This is a difficult question to assess in corals that occur over a wide depth-range, potentially throughout the euphotic zone. Thus, this study adds significantly to a research question for which there is a paucity of evidence. Deeper reefs - and marginal habitats of all types - have received increasing attention in a changing world and ocean, and so I believe this study will have a wide readership.

Dan Holstein

Reviewer #4 (Remarks to the Author):

This is a carefully designed study investigating coral dispersal in relation to parental habitats. The text is well written and the findings would be of most relevance to coral ecologists. I am less convinced the work would generate broader interest. This is due not to the technical execution of the work, but rather to its focus on brooding corals. There is also a very limited discussion of additional aspects of coral biology than may be contributing to the observations or their significance. Hence, I feel the interpretation of the results, particularly those aspects purported to have implications for reef building corals in general, is currently quite speculative.

The authors acknowledge that brooded corals can release fully developed planula larvae which may settle almost immediately or disperse widely. Those larvae inherit algal symbionts from their parents, which may predispose the symbiosis to optimum performance under the conditions where the parents have reproduced, or incur other costs to reconfigure their symbiont assemblages. Using a brooder of the same genus *Seriatopora*, Cooper et al (2011) demonstrated a change in Symbiodinium type between shallow and deep colonies and concluded that "photobiological flexibility is vital for persistence in contrasting light regimes, a shift in Symbiodinium type may also confer a functional advantage albeit at a metabolic cost with increased depth". This aspect of symbiotic assemblages may have very important consequences, as a multitude of species of microbes may be passed from parent corals to their brooded larvae, not just the dinoflagellates. "Coral host factors as well as the environmental bacterial pool play a role in shaping coral-associated bacterial community composition. Host factors may include microbe transmission mode (horizontal versus maternal) and host specificity" (Damjanovic et al (2019). The host cnidarian coral, the dinoflagellates and a mix of other microbes together produce the coral holobiont, but it is yet to be determined to what degree the holobiont is the unit of selection. Osman et al (2020) when investigating Symbiodiniaceae specificity, along with bacterial composition and diversity, in six Red Sea coral species, demonstrated the importance of identifying the holobiont consortium and its specificity. While the endosymbiotic community was conserved they concluded the dynamic composition of the corals' bacteria across sites may contribute to holobiont function and broaden the ecological niche. These studies raise many questions and admittedly involve different coral species and reef locations. They do, nonetheless, provide examples of endosymbiont plasticity with depth, variations in the level of vertical connectivity between reefs in different bioregions and the potentially important role of a broad range of microbe, that may come initially from parent colonies, in the fitness of the holobiont coral.

There is also the probability that the connectivity or otherwise between deep and shallow reef habitats

varies with species and locations. In some situations genetic connectivity between deep and shallow populations is not as common as horizontal connectivity, though demographically significant connectivity between deep and shallow habitats has been demonstrated. However, Van Oppen et al (2011), found with *Seriatopora* evidence for recruitment of larvae of deep water origin into shallow habitats on an Indian Ocean reef, but conversely, no migration from the genetically divergent deep slope populations into the shallow habitats on the Great Barrier Reef.

So, while this manuscript documents a good piece of science, I feel it is far too premature to read generic implications of parental effects for coral reefs into the results presented here. As such I recommend the authors rewrite, with less speculation.

Cooper Timothy F., Ulstrup Karin E., Dandan Sana S., Heyward Andrew J., Kühl Michael, Muirhead Andrew, O'Leary Rebecca A., Ziersen Bibi E. F. and Van Oppen Madeleine J. H. 2011 Niche specialization of reef-building corals in the mesophotic zone: metabolic trade-offs between divergent *Symbiodinium* types *Proc. R. Soc. B.* 278:1840–1850. <http://doi.org/10.1098/rspb.2010.2321>

Damjanovic et al (2019). Experimental Inoculation of Coral Recruits With Marine Bacteria Indicates Scope for Microbiome Manipulation in *Acropora tenuis* and *Platygyra daedalea*. *Front. Microbiol.*, <https://doi.org/10.3389/fmicb.2019.01702>

Osman, E. O., Suggett, D. J., Voolstra, C. R., Pettay, D. T., Clark, D. R., Pogoreutz, C., et al. (2020). Coral microbiome composition along the northern Red Sea suggests high plasticity of bacterial and specificity of endosymbiotic dinoflagellate communities. *Microbiome* 8, 1–16. doi: 10.1186/s40168-019-0776-5

Van Oppen et al 2011. The role of deep reefs in shallow reef recovery: an assessment of vertical connectivity in a brooding coral from west and east Australia. <https://www.doi.org/10.1111/j.1365-294X.2011.05050.x>

Shlesinger and Loya. Depth-dependent parental effects create invisible barriers to coral dispersal – response to referees

Reviewer #3

Comment 1: This is my second review of this manuscript, and the authors have made a careful set of revisions in response to my and the other reviewers' comments. As mentioned previously, while the central question of the study is not exactly novel - namely that habitat (in this case depth) structures the success of coral settlement and success, and that structuring may be reinforced through generational selection or adult fitness. This is a difficult question to assess in corals that occur over a wide depth-range, potentially throughout the euphotic zone. Thus, this study adds significantly to a research question for which there is a paucity of evidence. Deeper reefs - and marginal habitats of all types - have received increasing attention in a changing world and ocean, and so I believe this study will have a wide readership.

We thank the reviewer for reviewing our manuscript again, and for the appreciation and enthusiasm regarding our effort and main results. The reviewer comments on the earlier version of this manuscript were to-the-point and most helpful and for that, we are truly grateful.

Reviewer #4

Comment 1: This is a carefully designed study investigating coral dispersal in relation to parental habitats. The text is well written and the findings would be of most relevance to coral ecologists. I am less convinced the work would generate broader interest. This is due not to the technical execution of the work, but rather to its focus on brooding corals. There is also a very limited discussion of additional aspects of coral biology than may be contributing to the observations or their significance. Hence, I feel the interpretation of the results, particularly those aspects purported to have implications for reef building corals in general, is currently quite speculative.

We thank the reviewer for this summary and perspective of our manuscript. We have addressed the two main criticisms raised by the reviewer (i.e., discussion of additional aspects that may be contributing to our findings and the generality of our interpretations) below.

Comment 2: The authors acknowledge that brooded corals can release fully developed planula larvae which may settle almost immediately or disperse widely. Those larvae inherit algal symbionts from their parents, which may predispose the symbiosis to optimum performance under the conditions where the parents have reproduced, or incur other costs to reconfigure their symbiont assemblages. Using a brooder of the same genus *Seriatopora*, Cooper et al (2011) demonstrated a change in Symbiodinium type between shallow and deep colonies and concluded that “photobiological flexibility is vital for persistence in contrasting light regimes, a shift in Symbiodinium type may also confer a functional advantage albeit at a metabolic cost with increased depth”. This aspect of symbiotic assemblages may have very important consequences, as a multitude of species of microbes may be passed from parent corals to their brooded larvae,

not just the dinoflagellates. “Coral host factors as well as the environmental bacterial pool play a role in shaping coral-associated bacterial community composition. Host factors may include microbe transmission mode (horizontal versus maternal) and host specificity” (Damjanovic et al (2019). The host cnidarian coral, the dinoflagellates and a mix of other microbes together produce the coral holobiont, but it is yet to be determined to what degree the holobiont is the unit of selection. Osman et al (2020) when investigation Symbiodiniaceae specificity, along with bacterial composition and diversity, in six Red Sea coral species, demonstrated the importance of identifying the holobiont consortium and its specificity. While the endosymbiotic community was conserved they concluded the dynamic composition of the corals’ bacteria across sites may contribute to holobiont function and broaden the ecological niche. These studies raise many questions and admittedly involve different coral species and reef locations. They do, nonetheless, provide examples of endosymbiont plasticity with depth, variations in the level of vertical connectivity between reefs in different bioregions and the potentially important role of a broad range of microbe, that may come initially from parent colonies, in the fitness of the holobiont coral.

The reviewer raises here an excellent point. As we did not engage in describing the symbiont composition in our study, we initially felt that it would be overly speculative to develop much discussion around this issue. Nonetheless, we fully agree with the reviewer that this topic has relevance to our study, and we thank the reviewer for prompting us to address it. We have now included a brief discussion regarding the coral host-symbiont associations across depth, lines 224-237:

“In some species and localities, divergence of coral populations by depth might be further reinforced by specialized depth-specific coral host-symbiont associations (Cooper et al. 2011, van Oppen et al. 2011, 2018, Pochon et al. 2015, Serrano et al. 2016), which may also be related to the mode of larval development. Larvae of most brooding coral species directly inherit their dinoflagellate photosymbionts from their parent colonies (i.e., vertical transmission), while larvae or newly established polyps of many spawning corals acquire symbionts from the environment (i.e., horizontal transmission; Baird et al. 2009, Padilla-Gamiño et al. 2012). Thus, the parental effects we report here likely involve the mutual response of both the coral host and its endosymbiotic assemblage, which may further suggest that brooding corals predispose the symbiotic association of their offspring to optimal performance under parental habitat conditions. Whether our findings generally hold for corals as well as for other marine organisms, awaits similar assessments of broadcast-spawning species to be carried out. Yet, despite the natural variability between species and sites, genetic segregation of both the coral host and its symbionts across depth was found to date in both brooding and spawning corals (Eytan et al. 2009, Cooper et al. 2011, van Oppen et al. 2011, 2018, Prada and Hellberg 2013, Brazeau et al. 2013, Serrano et al. 2014, 2016, Pochon et al. 2015, Bongaerts et al. 2017, Eckert et al. 2019, Drury et al. 2020”

Comment 3: There is also the probability that the connectivity or otherwise between deep and shallow reef habitats varies with species and locations. In some situations genetic connectivity between deep and shallow populations is not as common as horizontal connectivity, though demographically significant connectivity between deep and shallow habitats has been demonstrated. However, Van Oppen et al (2011), found with *Seriatopora* evidence for recruitment of larvae of deep water origin into shallow habitats on an Indian Ocean reef, but

conversely, no migration from the genetically divergent deep slope populations into the shallow habitats on the Great Barrier Reef.

*We completely agree with the reviewer regarding the point raised here, which is why we clearly stated this in the introduction (lines 56-58) as follows: “Indeed, recent studies on coral population genetic structures have found evidence of segregation across depth (**although this may differ between sites among species**)”. As for the study by van Oppen et al. (2011) of both the coral host and its symbiont population genetic structure and connectivity in west and east Australia – being one of the earlier papers assessing these topics, we cite this paper in several places along our manuscript, and specifically after the statement that genetic structure of coral populations may differ between sites and among species.*

*Nonetheless, to further emphasize this point, we have now added a similar note in our revised discussion (lines 235-237), which reads “Yet, **although it might differ between species and sites**, genetic segregation across depth was found to date in both brooding and broadcast-spawning corals”*

Comment 4: So, while this manuscript documents a good piece of science, I feel it is far too premature to read generic implications of parental effects for coral reefs into the results presented here. As such I recommend the authors rewrite, with less speculation.

We have toned down the generalizations we had in the original manuscript and we now focus better on the take-home message. Additionally, we now further emphasize in several places in the manuscript that we worked with brooding corals, stressing that our results and hypotheses should be further assessed using broadcast-spawning organisms.

Below are the changes we made in response to this comment:

- 1. In the abstract, the following overly general sentence was deleted: “Here, we reveal one of the biological and ecological mechanisms that may explain the diverging population and community structures across depth.” Instead, we go straight to the methodological sentence and explicitly state that we studied brooding corals: “Here, we performed a series of ex-situ and in-situ experiments using coral larvae of three **brooding** species from contrasting shallow- and deep-water habitats...”*
- 2. In the final sentence of the abstract we removed the generalization “in the sea”.*
- 3. We have added the following sentence in the end of the introduction, which briefly summarizes our main findings and specifically address brooding corals (Lines 88-90): “Our results show that parental effects and biological responses during the early life-history stages of **brooding** corals impose selective constraints on their dispersal and settlement under different conditions, depending on their natal habitats.”*
- 4. We modified our conclusion in line 124 to be more specific to brooding corals, which now reads: “Our results further indicate that in **brooding** coral species inhabiting a wide depth gradient...”*
- 5. We toned down the conclusion stated in lines 194-195 and added “to some extent” so this sentence now reads as: “...and are structured, to some extent, in a niche- or habitat-dependent manner related to their natal environment”.*
- 6. The conclusion summarized in lines 212-215 was complemented by a specific emphasis of brooding corals as follows: “...demonstrate the possible invisible barriers imposed on larval dispersal of corals, **at least for brooding species**”.*

7. We added a specific reference to brooding corals in our conclusion in line 249, which now reads: “The parental effects and phenotype-environment mismatches that we found here in **brooding** corals...”
8. We added a specific reference to brooding corals in our conclusion in line 255, which now reads: “Here, we have shown that local adaptations and parental effects may considerably influence **brooding** coral abundance...”
9. We toned down the statement in line 259, which now reads: “which may explain **some of** the observed community and population zonation across depth.”
10. The reservation in lines 233-235: “Whether our findings generally hold for corals as well as for other marine organisms, awaits similar assessments of broadcast-spawning species to be carried out.” is now being echoed again in the final overall conclusion section in lines 263-264: “Accordingly, while similar studies using broadcast-spawning species are needed to assess the generality of the results we report here, our findings highlight that...”

Editor comments:

Comment 1: Following the new Reviewer 4's recommendations, we ask that you modify the language of your conclusions to generalize less to a wide range of corals in general and make clearer that these results are specific to brooding corals. We therefore invite you to revise your paper one last time to address the remaining concerns of this reviewer.

We have toned down the generalizations we had in the manuscript while focusing on the concluding statements, and also made it clearer in several places along the manuscript that we were studying brooding corals.

Below are all the relevant changes we made in response to this issue (same as the response to reviewer #4 comment 4):

1. *In the abstract, the following overly general sentence was deleted: “Here, we reveal one of the biological and ecological mechanisms that may explain the diverging population and community structures across depth.” Instead, we go straight to the methodological sentence and explicitly state that we studied brooding corals: “Here, we performed a series of ex-situ and in-situ experiments using coral larvae of three **brooding** species from contrasting shallow- and deep-water habitats...”*
2. *In the final sentence of the abstract we removed the generalization “in the sea”.*
3. *We have added the following sentence in the end of the introduction, which briefly summarize our main findings and specifically address brooding corals (Lines 88-90): “Our results show that parental effects and biological responses during the early life-history stages of **brooding** corals impose selective constraints on their dispersal and settlement under different conditions, depending on their natal habitats.”*
4. *We modified our conclusion in line 124 to be more specific to brooding corals, which now reads: “Our results further indicate that in **brooding** coral species inhabiting a wide depth gradient...”*

5. *We toned down the conclusion stated in lines 194-195 and added “to some extent” so this sentence now reads as: “...and are structured, to some extent, in a niche- or habitat-dependent manner related to their natal environment”.*
6. *The conclusion summarized in lines 212-215 was complemented by a specific emphasize of brooding corals as follows: “...demonstrate the possible invisible barriers imposed on larval dispersal of corals, **at least for brooding species**”.*
7. *We added a specific reference to brooding corals in our conclusion in line 249, which now reads: “The parental effects and phenotype-environment mismatches that we found here in **brooding** corals...”*
8. *We added a specific reference to brooding corals in our conclusion in line 255, which now reads: “Here, we have shown that local adaptations and parental effects may considerably influence **brooding** coral abundance...”*
9. *We toned down the statement in line 259, which now reads: “which may explain **some of** the observed community and population zonation across depth.”*
10. *The reservation in lines 233-235: “Whether our findings generally hold for corals as well as for other marine organisms, awaits similar assessments of broadcast-spawning species to be carried out.” is now being echoed again in the final overall conclusion section in lines 263-264: “Accordingly, while similar studies using broadcast-spawning species are needed to assess the generality of the results we report here, our findings highlight that...”*

Comment 2: At the same time we ask that you edit your manuscript to comply with our format requirements and to maximise the accessibility and therefore the impact of your work.

We edited our manuscript to comply with the journal requirements.